# Synthesis Routes on Electrochemical Behavior of Co-Free Layered LiNi_0.5_Mn_0.5_O_2_ Cathode for Li-Ion Batteries

**DOI:** 10.3390/molecules28020794

**Published:** 2023-01-13

**Authors:** Shu-Yi Tsai, Kuan-Zong Fung

**Affiliations:** 1Hierarchical Green-Energy Materials (Hi-GEM) Research Center, No. 1, Ta-Hsueh Road, Tainan 70101, Taiwan; 2Department of Materials Science and Engineering, National Cheng Kung University, No. 1, Ta-Hsueh Road, Tainan 70101, Taiwan

**Keywords:** LiNi_0.5_Mn_0.5_O_2_, Co-free, layer structure

## Abstract

Co-free layered LiNi_0.5_Mn_0.5_O_2_ has received considerable attention due to high theoretical capacity (280 mAh g^−1^) and low cost comparable than LiCoO_2_. The ability of nickel to be oxidized (Ni^2+^/Ni^3+^/Ni^4+^) acts as electrochemical active and has a low activation energy barrier, while the stability of Mn^4+^ provides a stable host structure. However, selection of appropriate preparation method and condition are critical to providing an ideal layered structure of LiNi_0.5_Mn_0.5_O_2_ with good electrochemical performance. In this study, Layered LiNi_0.5_Mn_0.5_O_2_ has been synthesized by sol-gel and solid-state routes. According to the XRD, the sol-gel method provides a pure phase, and solid-state process only minimize the secondary phases to certain limit. The Ni^2+^/Mn^4+^ content in the sol-gel process was higher than in the solid-state reaction, which may be due to the chemical composition homogeneity of the sol-gel samples. Regarding the electrochemical behavior, sol-gel process is better than solid-state reaction. The discharge capacity is 145 mAh/g and 91 mAh/g for the sol-gel process and solid-state reaction samples, respectively.

## 1. Introduction

High-capacity cathode materials typically contain a certain amount of Cobalt for stabilization and promoting their electrochemical properties. The role of cobalt as transition metal changes its oxidation state to maintain the electrically stay neutral when the lithium ion is taken out from the cathode. However, Cobalt price has gone up significantly so high that Co-free cathode materials have been proposed and investigated recently [1,2,3]. Layered-structure lithium nickel manganese oxide (LiNi_0.5_Mn_0.5_O_2_) is a candidate Co-free cathode material that possesses high theoretical capacity (280 mAh/g), good cycling stability, and small volume changes [4,5]. Ohzuku and Makimura successfully demonstrated the synthesis of 1:1 solid solution of LiNiO_2_ and LiMnO_2_, namely LiNi_0.5_Mn_0.5_O_2_ using solid-state synthesis technique by heating at 1000 °C for 15 h. LiNi_0.5_Mn_0.5_O_2_ is one of the most attractive materials [6].The structure of LiNi_0.5_Mn_0.5_O_2_ consists of layers of transition metal (Ni and Mn) separated from Li layers by oxygen. Li and transition metal (TM) coordinated octahedrally by oxygen but diffuses from site to site by hopping through intermediate tetrahedral sites. The Li migration during the charge-discharge process has a diffusion rate, which has an activation energy barrier. The energy required for a Li-ion to cross the activated state is likely to depend on the size of the tetrahedral site, which also calls as strain effect. However, the exchange of Ni/Li (or Ni/Li disordering) usually happens between the layer and TM layer in these materials during synthesis and electrochemical cycling [7]. Several amount of Li^+^ that occupy the transition metal slab or vice versa for transition metal and other indication of Li^+^/Ni^2+^ mixing is the Li_2_MnO_3_ like phase formation. The impact of Li^+^/Ni^2+^ disorder increases the possible formation of Li-dumbbells which causes it causes the high voltage process involving removal of the tetrahedral Li-ions [8].Furthermore, the transition metal of Ni acts as electrochemically active and has a low activation energy barrier due to the valence state is low, which cam promote the Li ion diffusion. On the other hand, manganese act as electrochemically passive, which the primary role is to maintain the host crystal stability. During synthesis layered LiNi_0.5_Mn_0.5_O_2_ has a possible cation anti-site contains approximately 8–10%, which mean it obtain 10% Li^+^ will occupy the transition metal slab and vice versa for transition metal [7]. This cation mixing promotes by two factor that must investigate during research, the first assumption caused by a Ni^2+^ substitution in the lithium layers due to the closely cationic size between Li^+^ (0.76 Å) and Ni^2+^ (0.69 Å). The second consideration is the formation of Li^+^/Mn^4+^ which form Li_2_MnO_3_-like phase ordering. Li_2_MnO_3_-like was suggested can provide a driving force for the Li^+^/Ni^2+^ exchange [9,10]. The selection of appropriate preparation method and condition are critical to obtain an ideal layered structure of LiNi_0.5_Mn_0.5_O_2_ showing good electrochemical performance. There are several methods for synthesis LiNi_0.5_Mn_0.5_O_2_ cathode material such as solid-state method [11], hydrothermal synthesis [12] and co-precipitation [13] have been used by other groups to synthesize this material. Furthermore, the precipitation agents require several purification steps to be removed and their residues can promote the negative impact for electrochemical performance. This difficulty when producing in term batch-to-batch during large-scale production. Nevertheless, the hydrothermal synthesized promoted the cation anti-site defect which determined electrochemical properties of cathode materials. Even worse, impurities have often been found along with hydrothermally due to the oxidizing circumstance in aqueous solution. To produce LiNi_0.5_Mn_0.5_O_2_ with a superior electrochemical performance, there still are challenges to be overcome. Sol-gel method is the common method for synthesis of the multi-cation cathode materials due to high purity, high homogeneity, and low synthesis temperatures. In this study, a sol-gel process is proposed to fabricate layered LiNi_0.5_Mn_0.5_O_2_ with good electrochemical performance. For comparison, LiNi_0.5_Mn_0.5_O_2_ was also synthesized by a conventional solid-state reaction. The samples were characterized by XRD, SEM, XPS and galvanostatic charge–discharge tests.

## 2. Experimental

### 2.1. Materials and Preparation

The sol-gel method preparation was prepared according to the procedure: 0.105 mole lithium acetate (Li(CH_3_COO) ·2H_2_O, MACKLIN, 99.9%) with an excess of 5 mol % to compensate for Li-loss during high temperature treatment, 0.05 mole nickel acetate (Ni(CH_3_COO)_2_·4H_2_O, Sigma Aldrich, St. Louis, MO, USA, 98 %), and 0.05 mole manganese acetate(Mn(CH_3_COO)_2_·4H_2_O, Sigma Aldrich, St. Louis, MO, USA, ≥99%)was dissolved in distilled water, and the same time the citric acid was dissolved with distilled water in the different beaker. The citric acid was added drop by drop in to the transition metal solution afterward adding the ethylene glycol. The temperature setting at 60–70 °C and stirring for gelation overnight. Furthermore, increased the temperature at 150 °C until getting a dry gel. The resulting dry gel was continued to pre-calcine in Al_2_O_3_ crucibles at 600 °C for 12 h and calcined at 900 °C for 12 h.

Solid-state method was used for comparing the sol-gel method. A stoichiometric amount of nickel oxide (NiO, Sigma Aldrich, St. Louis, MO, USA, 99.8%), manganese oxide (MnO_2_, Sigma Aldrich, St. Louis, MO, USA, ≥ 99%) and lithium hydroxide (LiOH, Sigma Aldrich, St. Louis, MO, USA, 98%) as raw materials. The raw materials were placed in a 50 mL ball mill jar to homogenized by roller milling for 24 h at 500 r.p.m using zirconia milling media with ethanol (Sigma Aldrich, St. Louis, MO, USA, 95%) added as a carrier fluid. The mass ratio of the starting materials and the zirconia balls was 1:90. After milling, ethanol was evaporated under mixing at 85 °C. After drying, each powder was sieved using a 325 mesh to obtain uniform particle sizes. Finally, the powders were calcined at 900 °C for 12 h to obtain LiNi_0.5_Mn_0.5_O_2_ powders.

### 2.2. Basic Characterization

The thermal behavior of the as-prepared powder was carried out using (SETSYS Evolution TGA-DTA/DSC SETARAM) up to 1000 °C at the scan rate of 20 °C/min. The phase purity and crystal structure of the LiNi_0.5_Mn_0.5_O_2_ samples were investigated using X-ray diffraction (Rigaku Multi Flex) with Cu-Kα radiation. The two thetas range from 10 to 80 degree with 0.5 deg./min rate. The operating voltage and current were 30 kV and 20 mA, respectively. Phase identification of the sample was analyzed using MDI Jade 6 and identify the phase using ICDD database. The date result calculated to determine the value of lattice parameter, average volume crystal size and amount of impurities phase. Scanning electron microscopy was performed using a Hitachi S3000 to identify the morphology and particle size of the samples. The particle size of powder synthesized was determined using Zetasizer 3000 HSA.X-ray photoelectron spectroscopy confirms the valence state of transition metal due to the important initial condition of nickel state and correction binding energy using C1s peak (285 eV).

### 2.3. Electrochemical Characterization

The electrodes were prepared by mixing the LiNi_0.5_Mn_0.5_O_2_ powder with carbon black and polyvinylidene fluoride at a weight ratio of 80:10:10 in N-methyl pyrrolidine. The slurry was coated on aluminum current collect using the doctor blade, keeping the thickness of the coated electrode around 25 µm and dried at 100 °C in vacuum for 24 h. The foils were rolled into thin sheets and cut into disks with a diameter of 13 mm. The cathode loading was estimated to be ∼2 mg/cm^2^. Lithium foil was used as the anode and polypropylene microporous films were used as separators. The electrolyte consisted of 1 M LiPF_6_ in a mixture of ethyl carbonate (EC, Sigma Aldrich, St. Louis, MO, USA) and diethyl carbonate (DMC, Sigma Aldrich, St. Louis, MO, USA) at a 1:1 volume ratio. CR2032 coin cells were assembled in an argon-filled glove box. The galvanostatic charge-discharge curves were performed using an Arbin Battery Tester 2043 in the potential range 2.5–4.3 V.

## 3. Results

### 3.1. Weight Loss Decomposition

Figure 1 shows the thermogravimetry and differential scanning calorimetry of the two-precursor powder for solid state method and the sol gel method. For solid state precursor, there show no big changes in weight loss curve. The high slope around 200–550 °C probably the weight of loss (17.30%) is attributed from thermal dehydration of LiOH.H_2_O and also CO_2_ releasing from carbonate decomposition [14]. Another weight-loss contribution also comes from MnO_2_ by releasing oxygen [15]. The weight loss above than 550 °C around (2.10%) probably attributed by formation phase compound such as spinel, rock-salt and layered phase. However, the theoretical and practical weight loss is slightly different, 19.73% and 24.75%, respectively. It is assumed the unseen of weight loss because of water vapor from LiOH.H_2_O. Attributing CO_2_ for kinetically can promote the vapor to occur at 60 °C [14]. This reason makes sense around 5.02% practical weight loss it does not appear.

For sol-gel precursor, in the begin observation at low temperature range up to 200 °C the weight loss (32.40%) has occurred, probably corresponding to the physically adsorbed water and weakly bound ligand molecules. The second stage of weight loss (42.28%) was occurred between 200–400 °C, that attributed to the pyrolysis of residential organic functional group such as glycerol in low temperature of this stage [16] and decomposition of acetate into oxides by accompanied releasing water by dehydration of vinyl alcohol (-CH-CHOH-) to leave (-CH=CH-) in the high-level temperature of this stage [17]. This phenomenon confirms by Nowak-wick et. al., the citric acid will decompose become an aconitic acid at 240 °C. The decomposition compound is a complex process leading through dehydration and decarboxylation reaction to different intermediate product [18]. The third stage shows a flat curve with small weight change about (1.49%), this weigh lost probably referred to the transformation of crystalline, rock-salt until had been formed the layered phase structure [19]. The weight loss of 2.1% between 500 and 950 °C has attributed to the release of O_2_.

### 3.2. X–ray Diffraction Analysis

The crystal formation during calcination was confirmed using X-ray diffraction based on temperature difference and synthesis routes process. Figure 2 demonstrate the phase transformation between sol-gel and solid-state does occur, respectively. The differences between sol-gel and solid state showed in the magnification (104) as shown in (b) and (e). The smoothed peak owns by sol-gel, indicate the layered structure well formed, while the solid state showed a broad region peak and accompany by the complex pattern with a few overlapping peaks. The complete splitting peak reflection from the layered phase can be clearly shown in (c) and (d). Those index (006)/(102) and (018)/(110) are considered to be indicator the well-organized of layered structure which mean the order distribution of lithium and transition metal ion in the lattice site. As increasing temperature, the enhances the splitting phenomenon.

### 3.3. Morphology

The sample prepared by sol-gel and solid-state were observed, there seems not to be a significant difference for the morphology, but slightly larger particle size for a solid-state sample as shown in Figure 3. There also observed the bigger particle size in solid-state, the agglomeration probably it is attributed by melted of particle together as shown in Figure 3e. This result is an agreement with the synthesis result from Kos et al., that the particles of samples synthesized by solid state shown larger sizes and are not as well separated compared to those from the sol-gel method [20]. This result consistent with the particle size distribution that shown in the Figure 3c,f the broad peak of sol-gel is narrower than solid state, indicates the sol-gel particle size is more homogeneous size and well distributed than solid-state.

### 3.4. XPS

The XPS spectra of Ni 2p3/2 for sol-gel and solid-state sample show in Figure 4a,b. The Ni 2p3/3 spectrum consists of the main peak and accompanies by broad satellite peak. The main peak could be assigned to the nickel ion with the divalent state. As can be seen from this figure, Binding Energies (BEs) from the main peak are located at 854.27 and 854.61 eV for sol-gel and solid-state sample, respectively. However, the peak position of BEs Ni 2p3/2 for sol-gel shifted to the lower BEs than solid-state sample which indicate the Ni^2+^ were dominant valence state than Ni^3+^. However, the main peak Mn2p_3/2_ position of solid-state showed a shift toward lower BEs than sol-gel, further the broad width-peak of sol-gel showed slightly larger than solid-state peak. This mean the Mn^4+^ were dominant for sol-gel. This result, also confirm by ratio of splitting peak. The total Mn^4+^ and LEP area divide Mn^3+^ are 61.64 and 56.21 for sol-gel and solid-state, respectively.

### 3.5. Electrochemical Performance

The galvanostatic investigation was performed on the Li/ LiNi_0.5_Mn_0.5_O_2_ cell assembled between sol-gel and solid-state sample calcine at 900 °C. The first charge-discharge curve record at 2.7 to 4.3 V under constant current 0.05 C, as shown in Figure 5. The illustration of the charge-discharge curve for sol-gel sample looks smooth and monotonous. Upon charge, the voltage curve steeply increases at 3.75 V followed by a lower slope as decreasing x-Li content in Li_x_Ni_0.5_Mn_0.5_O_2_, is clear that the sample consists of pure layered phase. From the charge, capacity indicates that 0.57 lithium can be removed from the layered Li_x_Ni_0.5_Mn_0.5_O phase below the voltage range 4.3 V. However, the discharge capacity of 145 mAh/g, corresponding with re-insertion of 0.51 lithium in the host crystal. According to the XRD and XPS, the sol-gel method provides a pure phase material and more completely forming hexagonal ordering crystal of layered LiNi_0.5_Mn_0.5_O_2._

To investigate the voltage fade would be better discus using the differential discharge capacity versus potential (dQ/dV). An anodic peak at 3.75 V which associated with the oxidation of Ni^2+^ to Ni^4+^ was showed in the Figure 6. It should be noted that the anodic peak of solid-state sample is slightly shifted towards a lower potential and its intensity is lower than that of the sol-gel, indicating a worse electrochemical activity and resulting a lower capacity of discharge platform in solid state result, which are perfectly coincident with the XPS observations.

Figure 7 shows the rate capability of the layered LiNi_0.5_Mn_0.5_O_2_ synthesized via the sol-gel method and the solid-state method. The capacity of the layered LiNi_0.5_Mn_0.5_O_2_ prepared by sol-gel method is higher than that solid-state method, and this difference increases under various rates increases. This result could be attributed to particles agglomerate resulting in the decrease in its specific surface area, which makes increases the Li diffusion path, so the deceased discharge capacity of the layered LiNi_0.5_Mn_0.5_O_2_ synthesized via the solid-state method. The capacity fading in solid state method is attributed to the average Mn valence is equal to or less than 3.5. The disordering cation leads to the cracking of particles and loss of electric contact while cycling.

## 4. Conclusions

Layer LiNi_0.5_Mn_0.5_O_2_ was successfully synthesized via sol-gel and solid-state routes. By weight loss measurement and X-Ray characterization recorded the decomposition product and phase obtain from room temperature until 950 °C. Relative pure phase can be obtained by the sol-gel method at low temperature, due to the short distance among lithium and transition metal formed in the precursor. Conversely, incompletely tree-phase transformation occurs on the solid-state method at the same temperature, indicate the deficient of energy. The larger particles from raw materials may need longer annealing time or higher temperature to complete the required reaction. Layer LiNi_0.5_Mn_0.5_O_2_ prepared by sol-gel delivered better electrochemical performance than solid-state method in terms of capacity fade and cycle life performance.

## Figures and Tables

**Figure 1 molecules-28-00794-f001:**
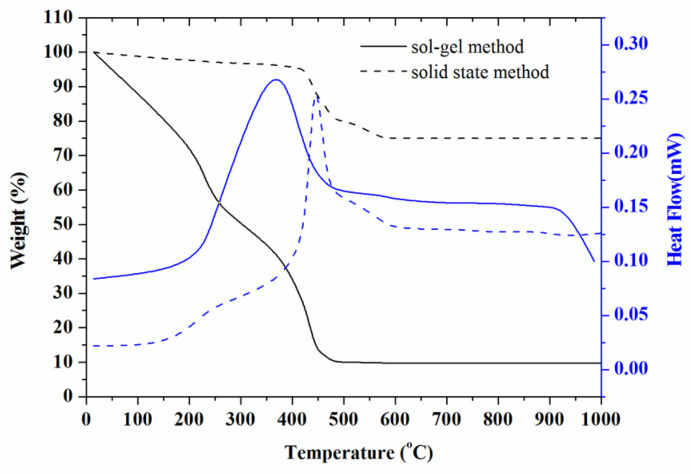
TG–DSC curves of the precursor powder prepared by sol-gel and solid state.

**Figure 2 molecules-28-00794-f002:**
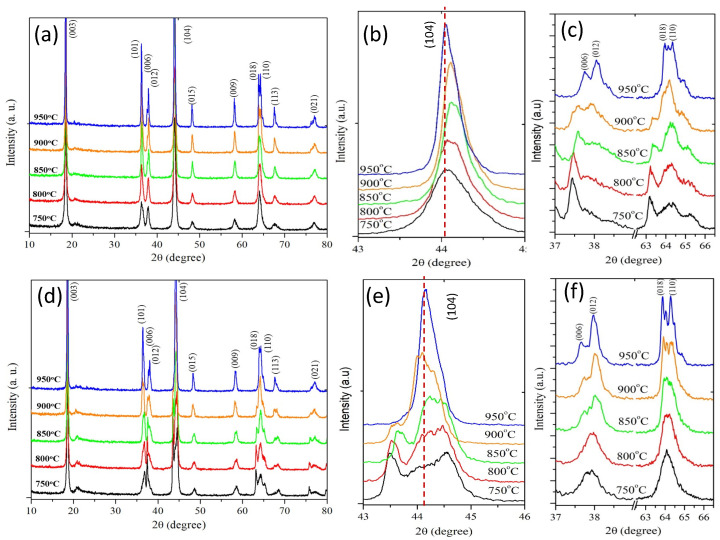
X-Ray pattern of LiNi_0.5_Mn_0.5_O_2_ sample prepared (**a**) by sol-gel sample and magnification of angle range (**b**) correspond (104) peak (**c**) splitting peak. (**d**) by solid-state sample and magnification of angle range (**e**) correspond (104) peak and (**f**) splitting peak.

**Figure 3 molecules-28-00794-f003:**
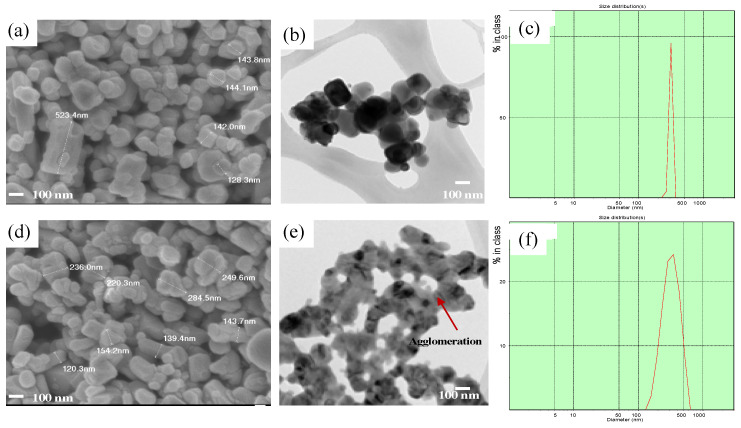
SEM morphology, TEM image and Particle size distribution of powder sample which calcine in the same temperature at 900 °C with different synthesis routes (**a**–**c**) sol-gel method and(**d**–**f**) solid state reaction method.

**Figure 4 molecules-28-00794-f004:**
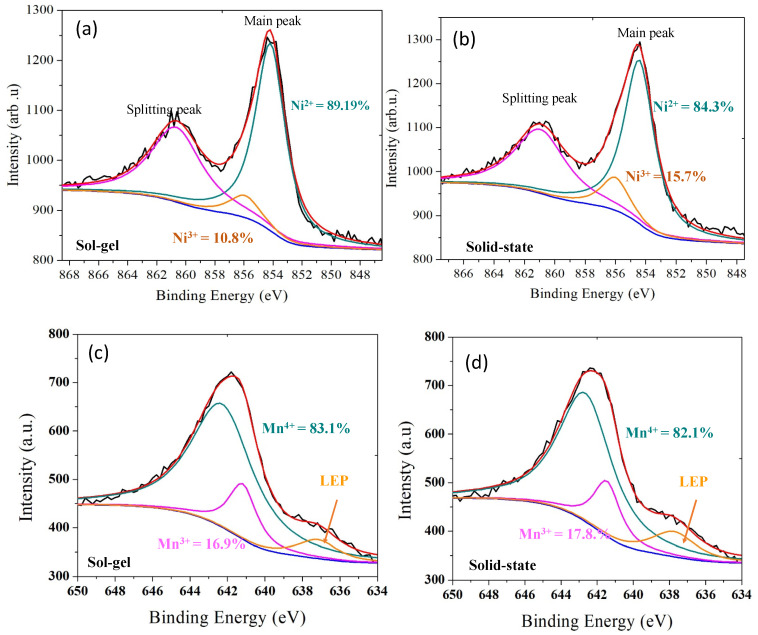
XPS spectra for the Ni2p_3/2_ photoemission line of the layered LiNi_0.5_Mn_0.5_O_2_ sample (**a**) Sol-gel and (**b**) Solid state. The Mn2p_3/2_ photoemission line of the layered LiNi_0.5_Mn_0.5_O_2_ sample (**c**) Sol-gel and (**d**) Solid state.

**Figure 5 molecules-28-00794-f005:**
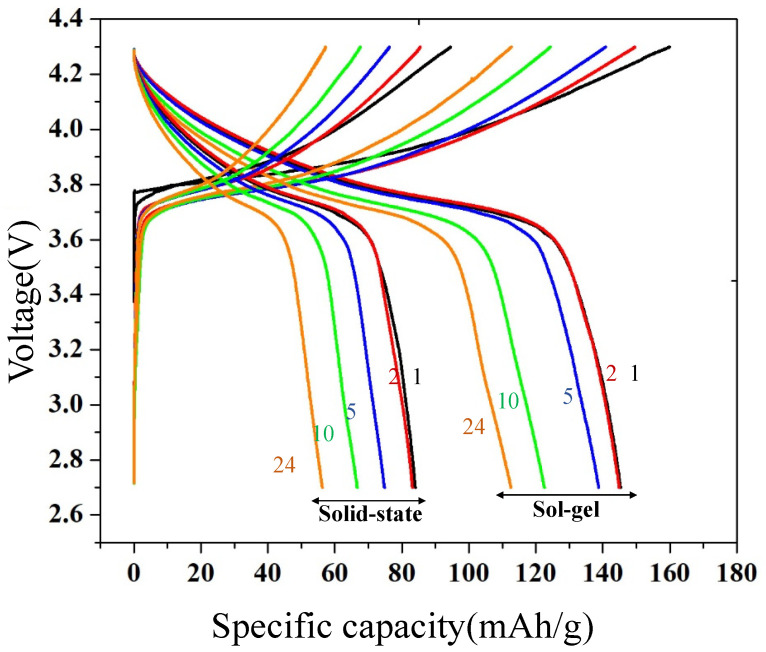
Electrochemical performance of layered LiNi_0.5_Mn_0.5_O_2_. First—fifth charge discharge voltage profile examines at 2.7-4.3 V under 0.05 C for sol-gel and solid-state samples.

**Figure 6 molecules-28-00794-f006:**
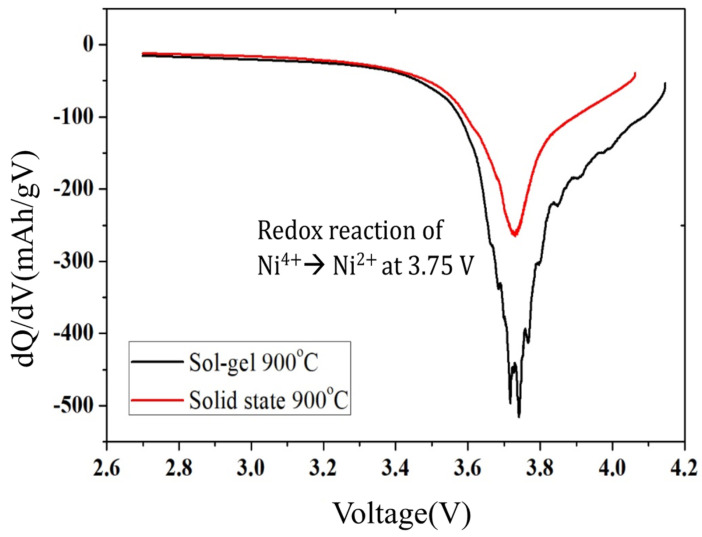
Differential capacity dQ/dV curve at the first cycle of sol-gel and solid-state method sample.

**Figure 7 molecules-28-00794-f007:**
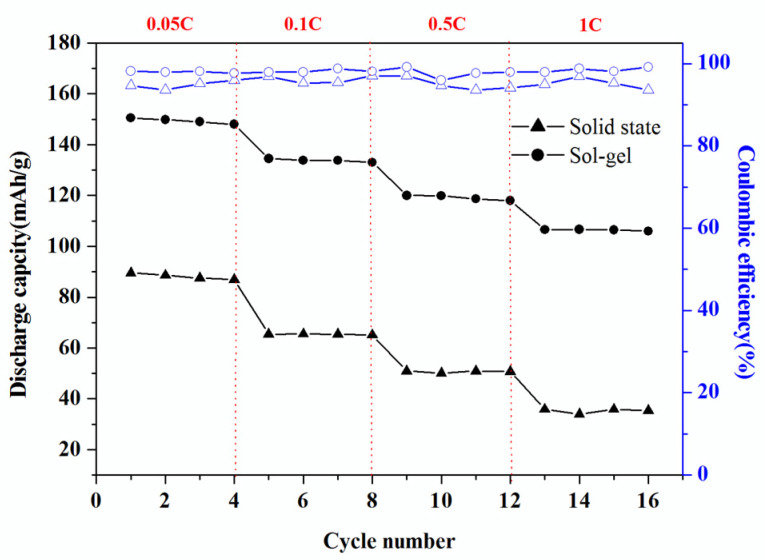
Rate capability of LiNi_0.5_Mn_0.5_O_2_ pellets synthesized by the sol-gel method and solid-state method at different C rates.

## Data Availability

Not applicable.

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
