# Peer review of "Synthesis Routes on Electrochemical Behavior of Co-Free Layered LiNi0.5Mn0.5O2 Cathode for Li-Ion Batteries"

_molecules, 2023, doi:10.3390/molecules28020794_

Round 1

Reviewer 1 Report

1. More experimental details should be provided, such as, the source and purity of reagents, excess ratio of lithium, 5%mol or more? For Solid state method, the volume of jar, ball-milling speed (xxx rpm), the volume of ethanol, and ratio of grinding media to material, etc. As for electrode preparation, such as the areal mass loading of active materials, and I cannot find the definition of 1C.

2. SEM images should be unified in the same magnification for comparing the size difference. And TEM images should be provided for a better clarification on the size difference.

3. For electrochemical performance, we can only see 5 cycles at 0.05C, more cycles should be provided, and cycles at various rates should be provided, besides, cycles at 1.0C or higher rate should be provided. Moreover, other results like CV, EIS spectra should be provided.

4. The language should be polished carefully, the reviewer can still find some grammar errors. And format of references should be amended,  information like pages is missing.

Author Response

Response to Reviewer 1 Comments

Point 1: More experimental details should be provided, such as, the source and purity of reagents, excess ratio of lithium, 5%mol or more? For Solid state method, the volume of jar, ball-milling speed (xxx rpm), the volume of ethanol, and ratio of grinding media to material, etc. As for electrode preparation, such as the areal mass loading of active materials, and I cannot find the definition of 1C.

Response 1: The experimental details have been corrected. The context has been revised as follows in line 77-96.

“The sol-gel method preparation was prepared according to the procedure: 0.105 mole lithium acetate (Li(CH3COO) ·2H2O, MACKLIN, 99.9%) with an excess of 5 mol % to compensate for Li-loss during high temperature treatment, 0.05 mole nickel acetate (Ni(CH3COO)2·4H2O, Sigma Aldrich,98 %), and 0.05 mole manganese acetate(Mn(CH3COO)2·4H2O ,Sigma Aldrich,≥99%)was dissolved in distilled water, and the same time the citric acid was dissolved with distilled water in the different beaker. . The citric acid was added drop by drop in to the transition metal solution afterward adding the ethylene glycol. The temperature setting at 60-70 °C and stirring for gelation overnight. Furthermore, increased the temperature at 150 °C until getting a dry gel. The resulting dry gel was continued to pre-calcine in Al2O3 crucibles at 600°C for 12 hours and calcined at 900°C for 12 hours.

Solid-state method was used for comparing the sol-gel method. A stoichiometric amount of nickel oxide(NiO,Sigma Aldrich,99.8%), manganese oxide(MnO2, Sigma Aldrich, ≥99%)and lithium hydroxide(LiOH,Sigma Aldrich,98%) as raw materials. The raw materials were placed in a 50 mL ball mill jar to homogenized by roller milling for 24 h at 500 r.p.m using zirconia milling media with ethanol (Sigma Aldrich,95%) added as a carrier fluid. The mass ratio of the starting materials and the zirconia balls was 1:90. After milling, ethanol was evaporated under mixing at 85°C. After drying, each powder was sieved using a 325 mesh to obtain uniform particle sizes. Finally, the powders were calcined at 900°C for 12h to obtain LiNi0.5Mn0.5O2 powders.”

.

Point 2: SEM images should be unified in the same magnification for comparing the size difference. And TEM images should be provided for a better clarification on the size difference.

Response 2: The suggested corrections are appreciated. SEM images in the same magnification,TEM and Particle size distribution diagram result has been added in Figure 3.

In response to the referee’s comment, the manuscript has been modified as follows.

“The sample prepared by sol-gel and solid-state were observed, there seems not to be a significant difference for the morphology, but slightly larger particle size for a solid-state sample as shown in Figure 3. There also observed the bigger particle size in solid-state, the agglomeration probably it is attributed by melted of particle together as shown in Fig(e). This result is an agreement with the synthesis result from Kos et al., that the particles of samples synthesized by solid state shown larger sizes and are not as well separated compared to those from the sol-gel method[18]. This result consistent with the particle size distribution that shown in the Fig. 3(c) and (f) the broad peak of sol-gel is narrower than solid state, indicates the sol-gel particle size is more homogeneous size and well distributed than solid-state””

Point 3: For electrochemical performance, we can only see 5 cycles at 0.05C, more cycles should be provided, and cycles at various rates should be provided, besides, cycles at 1.0C or higher rate should be provided. Moreover, other results like CV, EIS spectra should be provided..

Response 3: The effects of rate capability at various rates are considered. The capacity of the layered LiNi0.5Mn0.5O2 prepared by sol-gel method is higher than that solid-state method, and this difference increases under various rates increases as shown in Figure 7.The context has been revised as follows in line 219-227.

“Figure 7 shows the rate capability of the layered LiNi0.5Mn0.5O2 synthesized via the sol-gel method and the solid-state method. The capacity of the layered LiNi0.5Mn0.5O2 prepared by sol-gel method is higher than that solid-state method, and this difference increases under various rates increases. This result could be attributed to particles agglomerate resulting in the decrease of its specific surface area, which makes increases the Li diffusion path, so the deceased discharge capacity of the layered LiNi0.5Mn0.5O2 synthesized via the solid-state method. The capacity fading in solid state method is attributed to the average Mn valence is equal to or less than 3.5. The disordering cation leads to the cracking of particles and loss of electric contact while cycling.”

Point 4: he language should be polished carefully, the reviewer can still find some grammar errors. And format of references should be amended, information like pages is missing.

Response 4: The references format has been corrected. The manuscript has been corrected and polished by an English proficient professional.

Along with this letter, we are attaching the revised paper and all of the review materials. We expect this manuscript can meet the publication policy and publication quality of the journal. Please direct further correspondence and thank you very much for your kind consideration. If there is anything further we should do concerning this paper, please let us know. We can be contacted as follows.

Shu-Yi Tsai

Hierarchical Green-Energy Materials (Hi-GEM) Research Center,

National Cheng Kung University,

Tainan, TaiwanFax: +886-6-2380208

Reviewer 2 Report

In this manuscript, the author reports a sol-gel method to prepare LiNi0.5Mn0.5O2. The as-synthesized LiNi0.5Mn0.5O2 cathode displays excellent electrochemical performance. The mechanism for the excellent electrochemical of LiNi0.5Mn0.5O2 is revealed as well. The research of theory is sufficient and reasonable, accompanied by rigorous logic and clear expression. Based on its systematisms and significance, I would like to recommend its publication in Molecules after addressing the following issues.

1. In order to better reveal the changes of mass and heat, it is highly suggested to provide TG-DSC curve of precursor heating.

2. Particle size distribution diagram is recommended to add in Fig3.

3. The cycle performances of LiNi0.5Mn0.5O2 prepared by sol-gel and solid-state method should be provided.

4. The references format should be further proofread, such as subscript.

Author Response

Response to Reviewer 2 Comments

Point 1: In order to better reveal the changes of mass and heat, it is highly suggested to provide TG-DSC curve of precursor heating.

Response 1: We have transformed weight loss into TG-DSC as shown in Fig.1.

Point 2: Particle size distribution diagram is recommended to add in Fig3.

Response 2: The suggested corrections are appreciated. SEM images in the same magnification,TEM and Particle size distribution diagram result has been added in Figure 3.

In response to the referee’s comment, the manuscript has been modified as follows.

“The sample prepared by sol-gel and solid-state were observed, there seems not to be a significant difference for the morphology, but slightly larger particle size for a solid-state sample as shown in Figure 3. There also observed the bigger particle size in solid-state, the agglomeration probably it is attributed by melted of particle together as shown in Fig(e). This result is an agreement with the synthesis result from Kos et al., that the particles of samples synthesized by solid state shown larger sizes and are not as well separated compared to those from the sol-gel method[18]. This result consistent with the particle size distribution that shown in the Fig. 3(c) and (f) the broad peak of sol-gel is narrower than solid state, indicates the sol-gel particle size is more homogeneous size and well distributed than solid-state””

Point 3: The cycle performances of LiNi0.5Mn0.5O2 prepared by sol-gel and solid-state method should be provided.

Response 3: The effects of rate capability at various rates are considered. The capacity of the layered LiNi0.5Mn0.5O2 prepared by sol-gel method is higher than that solid-state method, and this difference increases under various rates increases as shown in Figure 7.The context has been revised as follows in line 219-227.

“Figure 7 shows the rate capability of the layered LiNi0.5Mn0.5O2 synthesized via the sol-gel method and the solid-state method. The capacity of the layered LiNi0.5Mn0.5O2 prepared by sol-gel method is higher than that solid-state method, and this difference increases under various rates increases. This result could be attributed to particles agglomerate resulting in the decrease of its specific surface area, which makes increases the Li diffusion path, so the deceased discharge capacity of the layered LiNi0.5Mn0.5O2 synthesized via the solid-state method. The capacity fading in solid state method is attributed to the average Mn valence is equal to or less than 3.5. The disordering cation leads to the cracking of particles and loss of electric contact while cycling.”

Point 4: The references format should be further proofread, such as subscrip

Response 4: The references format has been corrected.

Reviewer 3 Report

The authors studied the effect of synthesis routes on electrochemical behavior of Co-free layered LiNi0.5Mn0.5O2 cathode for Li-ion batteries. The paper provides some guides for the researchers. However, the manuscript should be carefully revised before consideration for publication.

1. The abstract should be revised. The author should highlight the novelty and main results of their research.

2. In the Introduction section, more related previous works should be cited to enrich the background.

3. Title of section 3 is wrong.

4. TEM images are suggested to provide.

5. More electrochemical performance data should be provided, including the cycling performance of more cycle times, rate performance, CV curves and EIS analysis.

6. The reason for the difference in performance of the materials prepared by different methods should be well discussed.

Author Response

Response to Reviewer 3 Comments

Point 1: The abstract should be revised. The author should highlight the novelty and main results of their research.

Response 1: The abstract has been modified. In response to the referee’s comment, the manuscript has been modified as follows.

“Co-free layered LiNi0.5Mn0.5O2 has received considerable attention due to high theoretical capacity (280 mAh g-1) and low cost comparable than LiCoO2. The ability of nickel to be oxidized (Ni2+/Ni3+/Ni4+) acts as electrochemical active and has a low activation energy barrier, while the stability of Mn4+ provides a stable host structure. However, selection of appropriate preparation method and condition are critical to providing an ideal layered structure of LiNi0.5Mn0.5O2 with good electrochemical performance. In this study, Layered LiNi0.5Mn0.5O2 has been synthesized by sol-gel and solid-state routes. According to the XRD, the sol-gel method provides a pure phase, and solid-state process only minimize the secondary phases to certain limit. The Ni2+/Mn4+ content in the sol-gel process was higher than in the solid-state reaction, which may be due to the chemical composition homogeneity of the sol-gel samples. Regarding the electrochemical behavior, sol-gel process is better than solid-state reaction. The discharge capacity is 145 mAh/g and 91 mAh/g for the sol-gel process and solid-state reaction samples, respectively.”

Point 2: In the Introduction section, more related previous works should be cited to enrich the background

Response 2: We have corrected this error and added some works into the references of this paper, in order to appropriately describe the versatile development of this subject. In response to the referee’s comment, the Introduction has added as follows. The context has been revised as follows in line 50-57.

“During synthesis layered LiNi0.5Mn0.5O2has a possible cation anti-site contains approximately 8-10%, which mean it obtain 10% Li+ will occupy the transition metal slab and vice versa for transition metal[7]. This cation mixing promotes by two factor that has to investigate during research, the first assumption caused by a Ni2+ substitution in the lithium layers due to the closely cationic size between Li+ (0.76 Å) and Ni2+ (0.69 Å). The second consideration is the formation of Li+/Mn4+ which form Li2MnO3-like phase ordering. Li2MnO3-like was suggested can provide a driving force for the Li+/Ni2+ exchange[9, 10].”

Point 3: Title of section 3 is wrong

Response 3: The Title of section has been corrected.

Point 4: TEM images are suggested to provide

Response 4: The suggested corrections are appreciated. SEM images in the same magnification,TEM and Particle size distribution diagram result has been added in Figure 3..

Point 5: More electrochemical performance data should be provided, including the cycling performance of more cycle times, rate performance, CV curves and EIS analysis.

Response 5: The effects of rate capability at various rates are considered. The capacity of the layered LiNi0.5Mn0.5O2 prepared by sol-gel method is higher than that solid-state method, and this difference increases under various rates increases as shown in Figure 7.The context has been revised as follows in line 219-227.

“Figure 7 shows the rate capability of the layered LiNi0.5Mn0.5O2 synthesized via the sol-gel method and the solid-state method. The capacity of the layered LiNi0.5Mn0.5O2 prepared by sol-gel method is higher than that solid-state method, and this difference increases under various rates increases. This result could be attributed to particles agglomerate resulting in the decrease of its specific surface area, which makes increases the Li diffusion path, so the deceased discharge capacity of the layered LiNi0.5Mn0.5O2 synthesized via the solid-state method. The capacity fading in solid state method is attributed to the average Mn valence is equal to or less than 3.5. The disordering cation leads to the cracking of particles and loss of electric contact while cycling.”

Point 6: The reason for the difference in performance of the materials prepared by different methods should be well discussed.

Response 6: The impact of Li+/Ni2+ disorder increases the possible formation of Li-dumbbells which causes it causes the high voltage process involving removal of the tetrahedral Li-ions. The advantages of the sol-gel process for mixing the metals ion at the atomic scale is providing a homogeneous particle composition. Furthermore, the sol-gel process provides narrow particle size distribution and lower annealing temperature by highly reactive powder reaction than conventional solid-state reaction. Furthermore, the precipitation agents require several purification steps to be removed and their residues can promote the negative impact for electrochemical performance. This difficulty when producing in term batch-to-batch during large-scale production. Nevertheless, the hydrothermal synthesized promoted the cati2on anti-site defect which determined electrochemical properties of cathode materials. Even worse, impurities have often been found along with hydrothermally due to the oxidizing circumstance in aqueous solution. In this study, a sol-gel process is proposed to fabricate layered LiNi0.5Mn0.5O2 with good electrochemical performance.

Along with this letter, we are attaching the revised paper and all of the review materials. We expect this manuscript can meet the publication policy and publication quality of the journal. Please direct further correspondence and thank you very much for your kind consideration. If there is anything further we should do concerning this paper, please let us know. We can be contacted as follows.

Shu-Yi Tsai

Hierarchical Green-Energy Materials (Hi-GEM) Research Center,

National Cheng Kung University,

Tainan, TaiwanFax: +886-6-2380208

Round 2

Reviewer 1 Report

The authors have well addressed my comments. I'd be glad to recommend it for publication in its current form.

Reviewer 3 Report

The manuscript has been improved after revision and can be accepted in its present form.